# SiC-Coated Carbon Nanotubes with Enhanced Oxidation Resistance and Stable Dielectric Properties

**DOI:** 10.3390/ma14112770

**Published:** 2021-05-24

**Authors:** Rong Li, Yuchang Qing, Juanjuan Zhao, Shiwen Huang

**Affiliations:** 1Xi’an Research Institute of High Technology, Xi’an 710025, China; xss760829@163.com; 2State Key Laboratory of Solidification Processing, School of Materials Science and Engineering, Northwestern Polytechnical University, Xi’an 710072, China; nwpu520@163.com (J.Z.); 15190853693@163.com (S.H.)

**Keywords:** carbon nanotubes, polycarbosilane, pyrolysis, antioxidation, dielectric properties

## Abstract

Carbon nanotubes (CNTs) coated with SiC coating was successfully prepared by pyrolysis of polycarbosilane (PCS) used as a precursor. The function of pyrolysis temperature on the oxidation resistance and the dielectric properties of CNTs/SiC were studied in X-band. The results demonstrate that the obtained dense SiC film can prevent the oxidation of CNTs when the pyrolysis temperature reaches 600 °C. Correspondingly, after heat treatment is at 400 °C for 200 h, the mass loss of P-600 is less than 1.86%, and the real and imaginary parts of the dielectric constant nearly keep constant (ε′ from 14.2 to 14, and ε″ from 5.7 to 5.5). SiC-coated CNTs have a better oxidation resistance than pristine CNTs. Therefore, this work, with a facile preparation process, enhances the oxidation resistance of CNTs at high temperature for a long time and maintains a stable dielectric property, which means CNTs/SiC composites can be good candidates for applications in the field of high-temperature absorbers.

## 1. Introduction

Electromagnetic (EM) waves have been extensively used in various fields, including wireless communications, analytical equipment, and radar technology, which in turn causes serious EM pollution [1,2,3,4,5,6]. In order to solve this problem, the preparation of excellent EM shielding materials has been focused in civil and military fields [7,8,9,10]. Nowadays, exceptional microwave absorbers with the merits of thin thickness, light weight, and strong EM wave absorption in a wide frequency range are urgently required [11,12].

In recent years, carbon nanotubes (CNTs), firstly discovered by Iijima in 1991, have become the most attractive absorbing material in terms of wide band and having a considerable microwave absorption capability due to its low density, high conductivity, large specific surface area, excellent mechanical properties, and unique EM characteristics [13,14,15,16,17]. However, CNTs are easily oxidized at high temperature, which has limited the practical application of CNTs in the field of high-temperature EM waves absorption [18,19].

At present, silicide ceramics as a series of ceramic coating, such as SiC, Si_3_N_4_, and MoSi_2_, have been widely researched in the anti-oxidation coating system. According to the previous reports, SiC can be produced through high-temperature reaction, which is used to fill defects such as cracks in the coating or as a sealing material to prevent the infiltration of oxygen. Because the oxygen diffusion coefficient of SiC is very low (less than 10^−13^ g/(cm·s), at 1200 °C), it can effectively provide protection for CNTs [20,21,22]. To date, SiC coatings have been successfully fabricated by many methods, such as physical vapor deposition (PVD) [23], chemical vapor deposition (CVD) [24], and vacuum plasma spraying technology [25]. However, these methods require long production time and expensive equipment and are heavily environment-dependent. Compared with the above methods, the polymer derivatization method, with the advantages of short production time, simple equipment requirements, easily controlled structure, and environmental protection, has been successfully used to fabricate SiC coating by pyrolysis of silicon containing a polymer precursor [26,27,28].

In addition, polycarbosilane (PCS), as an important precursor, has been effectively applied for the preparation of SiC coatings [28]. For example, Gupta et al. [29] show that PCS is chosen to generate SiC coating on CNTs, which can obviously improve the thermal oxidation stability and mechanical properties of CNTs. Luo et al. [30] have synthesized MWCNTs/SiC_x_O_y_ composite through the high temperature pyrolysis process, and the results demonstrated that the concentration of PCS had an important influence on the antioxidant properties of the coated MWCNTs. However, the experiment procedure is very difficult due to the higher preparation temperature of SiC coating. At the same time, there are rare research projects carried out about the oxidation resistance of CNTs from the perspective of oxidation duration and dielectric properties at high temperature.

In this work, we investigate the oxidation resistance and dielectric property of PCS-coated CNTs pyrolyzed at different temperatures (400 °C, 600 °C, and 800 °C). Furthermore, the oxidation resistance and complex permittivity in X-band of SiC/CNTs composites after heat treatment at 400 °C, for 200 h, were also investigated. The remarkable oxidation resistance and dielectric properties of SiC/CNTs obtained in high temperature could be a good candidate for preparing high-temperature radar absorbing materials.

## 2. Experimental

### 2.1. Materials

The PCS (—[Si-HCH_3_CH_2_]_n_—) with a low melting point (180~210 °C), molecular weight (1800~2000), and oxygen content (≤1.0 wt.%) was bought from Suzhou Sailifei Ceramic Fiber Co., Ltd. CNTs with the merits of diameter (7–15 nm), length (~10 μm), purity (>mass 97%), and ash (<mass 3%) were provided by Shenzhen Nanotech Port Co. Ltd., Shenzhen, China.

### 2.2. Samples Preparation

Firstly, PCS and xylene as the solvent were mixed and sonicated for 20 min. Subsequently, CNTs were added in the above solution by ultra-sonicating for 6 h. The slurry was placed in a water bath and then sieved to obtain a homogeneously dispersed powder. The PCS-coated CNTs powder (PCS: CNTs = 1:5) was pyrolyzed in a graphite crucible under vacuum furnace (9.6 × 10^−3^ Pa) with a heating rate of 10 °C/min and held at 400 °C, 600 °C, and 800 °C for 0–200 h, respectively. In addition, the obtained different SiC-coated CNTs powders were further heat treated at 400 °C, for 200 h, in air. For convenience, the PCS-coated CNTs without pyrolysis were marked as P-N, and the PCS-coated CNTs powder by pyrolysis, at 400 °C, 600 °C, and 800 °C, was denoted as P-400, P-600, and P-800, respectively.

### 2.3. Measurements

To study the microstructure transformation of SiC coatings at different pyrolysis temperatures, mass fractions of PCS powders were measured by a Thermogravimetry-differential scanning calorimetry (TG-DSC, SDT Q600, TA Instruments, Chicago, IL, USA) under N_2_, at 10 °C/min, and the molecular structure transformation of PCS powders was obtained by Fourier transform infrared spectrometer (FTIR, Nicolet is10, Thermo, Waltham, MA USA), in the range of 2500–400 cm^−1^. The crystal structure and micromorphology of these materials were monitored via X-ray diffraction (XRD, Philips, Eindhoven, The Netherlands) and scanning electron microscopy (SEM, Model JSM-6360, JEOL, Tokyo, Japan), respectively. The pre-measured samples were prepared by mixing with epoxy, with a filler ration of 10 wt.%, and the complex permittivity of SiC-coated CNTs was investigated using a network analyzer (Agilent technologies E8362B) in the X-band.

## 3. Results and Discussion

### 3.1. Dielectric Properties of CNTs Annealed at 400 °C with Different Time

In general, the EM wave absorption performances of dielectric absorbers are calculated by complex permittivity (ε_r_ = ε′ − jε″) [31]. Here, the real permittivity (ε′) and the imaginary permittivity (ε″) represent the storage capability and loss capacity of EM energy, respectively. In order to evaluate the oxidation resistance properties of CNTs after high-temperature heat treatment, the complex permittivity of 10 wt.% CNTs/wax was measured in the X-band. The ε′ and ε″ values of the obtained samples annealed at 400 °C with different time of 0–6 h are shown in Figure 1a,b, respectively. As the annealed time increased, the ε′ and ε″ values of the obtained CNTs gradually decreased, which can be explained by two factors, namely, relaxation phenomenon and electrical conductivity, as shown in Equation (1) [2]:(1)$$\varepsilon =\;\varepsilon_{\mathrm{relax}}\prime \prime \;+\;\sigma /\omega \varepsilon_{0}$$

Here, ε_relax_″ and ε_0_ are the electron relaxation process and dielectric constant in a vacuum, respectively. *σ* and *ω* are, respectively, electrical conductivity and the angular frequency. Based on that, the electrical conductivity of CNTs decreases with the extension of oxidation time, thereby resulting in the gradually reduced ε″ value. Interestingly, the curves of ε′ and ε″ emerge two vibration peaks when the oxidation time is 4 h, which is the so-called nonlinear phenomenon related to space charge polarization, dipole polarization, and interfacial polarization [32]. This phenomenon shows that the oxidation resistance and dielectric properties of CNTs at high temperature need to be further improved. From the above discussion, it can be concluded that the dielectric properties of CNTs, after being annealed at 400 °C, have a great decrease due to the presence of high temperature oxidation; therefore, it is essential to perform oxidation resistance treatment on CNTs for improving their high-temperature dielectric properties.

### 3.2. The Pyrolysis Mechanism and Products of PCS at Different Temperatures

Figure 2a shows the TG-DSC curves of PCS with a heating rate of 10 °C/min, under flow of N_2_ gas. The mass loss of PCS is about 36% when the pyrolysis temperature increases to 1100 °C [30,33]. The TGA curve could be separated into three stages: Firstly, when the temperature is below 385 °C, the main mass loss is due to the thermal degradation of PCS and the evaporation of small molecular oligomers [34]. Secondly, when the temperature increases to 708 °C, the mass loss of 19.1% may be caused by the cleavage of the side chains (-H_2_ and -CH_4_) of the PCS. In addition, the conversion of PCS from polymer into inorganic structure is also occurred in this stage. Thirdly, when the temperature is above 708 °C, the weight loss is mainly due to the elimination of residual hydrogen and the complete conversion of PCS from polymer to ceramic. Furthermore, the red curve, indicating DSC, shows four exothermic peaks at 230, 501, 794, and 1062 °C, respectively. Additionally, the small exothermic peak at 230 °C may be derived from the melting of PCS [35]. A broad peak at the region of 576–794 °C demonstrates the transformation of PCS from organic to inorganic. Moreover, the exothermic behavior above 794 °C indicates the transition from amorphous SiC to crystalline β-SiC [36]. The results demonstrated that the amorphous and dense anti-oxidation performance of PCS is due to the generation of crystalline and discontinuous SiC.

The characteristic absorption peaks of the functional group for PCS pyrolyzed at high temperature are indicated by the FT-IR spectra, as shown in Figure 2b. The two strong absorption peaks around 2120 cm^−1^ and 940 cm^−1^ are attributed to the presence of stretching vibration and bending vibration of the Si-H functional group, respectively. The absorption peaks around 1250 cm^−1^ and 1630 cm^−1^ are attributed to stretching vibrations and C=C vibration in Si-CH_3_ [37]. Moreover, the peak appeared at 1000 cm^−1^ can be explained by CH_2_ vibration in the Si-CH_2_-Si chain [38]. Furthermore, the intensity of the absorption peaks for Si-CH_3_ (1250 cm^−1^) and Si-H (2120 cm^−1^) is significantly decreased as the pyrolysis temperature rises from room temperature to 600 °C [33,39]. When the pyrolysis temperature is up to 800 °C, the Si-CH_3_ and Si-H functional groups are completely disappeared. Combined with the TG-DSC curves of PCS in the range of 400~600 °C, the broken Si-H bond may be due to the dehydrogenation reaction-generated H_2_ and small molecule oligomers [40]. However, since the samples began to transform from organic to inorganic, the transformation was not complete in these stages, even though it formed an amorphous SiC. When the temperature reaches 800 °C, crystalline β-SiC are formed by the transformation of amorphous SiC [41].

The evolution of XRD patterns for the samples with various pyrolysis temperatures is shown in Figure 3. The product pyrolyzed at 400 °C shows a broad peak at 2θ = 35°, corresponding to the (111) peak of β-SiC. Based on the previous analysis, there was a wide diffraction peak at 35° for PCS pyrolysis products at 400 °C, which may be related to the formation of SiC crystallites. The structure of the pyrolysis product was mainly amorphous, which indicated the intermediate transition process from organic to inorganic transformation, at temperatures lower than 400 °C. At 400 °C, the pyrolysis was not complete, and the products were mostly amorphous. When the temperature rose to 600 °C, the (111) peak became obvious, and the weak diffraction peaks at 60° and 72° were visible, respectively. At 600 °C, the diffraction peaks were low and wide, indicating that the pyrolysis products are mainly microcrystalline. As the pyrolysis temperature increased, the (111) peak gradually shifted and the intensity increased. Moreover, the diffraction peaks (220) and (330) of β-SiC appeared when the temperature was higher than 600 °C. Therefore, when the pyrolysis temperature is 600 °C, the pyrolysis products of PCS are microcrystalline, leading to the formation of the dense SiC coating. With the increase of temperature to 800 °C, the diffraction peak intensity of the pyrolysis products increases and the half-peak width decreases. According to the Scherrer equation, the SiC grain size continues to increase with increasing temperature [33]. Meanwhile, the gradual grain growth, continuity, and integrity of SiC coating are destroyed.

SEM images of PCS-coated CNTs after heat treatment at different temperatures are shown in Figure 4. When the pyrolysis temperature rises to 400 °C, CNTs of sample P-400 are tightly intertwined with each other and the surface is smooth (Figure 4a). For sample P-600, the SiC grains are fine and have good continuity between each other when coated with CNTs (Figure 4b). However, numerous isolated SiC grains are formed and detached when the pyrolysis temperature increases to 800 °C (Figure 4c). Meanwhile, the surface of sample P-800 becomes gradually rougher, and the grains become more independent because of the increasing pyrolysis temperature. As a result, the compactness of SiC coating is weakened, which in turn reduces the ability to prevent oxygen permeation. The formation of PCS-coated CNTs pyrolyzed at 600 °C is further confirmed by the presence of Si and C element peaks in EDS analysis (Figure 4d,e) [29].

Figure 4f indicates the mass loss curves of different pyrolysis temperatures (400 °C, 600 °C, and 800 °C) and pyrolysis times (0–200 h) on the oxidation resistance of PCS-coated CNTs powders. The oxidation mass loss rate of pyrolysis products at different temperatures varies greatly. Especially, the mass losses of sample P-N and P-400 are 75% and 62%, respectively. These results may be explained by the presence of continuous appearance and the amorphous SiC film. It is worth mentioning that the mass loss of sample P-600 is only 1.9%, which is because the size of SiC crystal obtained at 600 °C is relatively small and prevents the internal oxidation of particles. Therefore, the mass of the sample decreased quite slowly. However, for samples P-800, the mass of SiC-coated CNTs powders decreased to 59% because of the higher SiC content and the larger grain size, resulting in the fact that the SiO_2_ film could not be formed after pyrolysis. From the above analysis, it can be concluded that PCS-coated CNTs powders, pyrolyzed at 600 °C, possess the best antioxidant capacity, and the existence of SiC coating effectively prevents the corrosion of CNTs by oxygen and increases the life of CNTs in an aerobic environment.

### 3.3. Dielectric Properties of SiC-Coated CNTs

Figure 5a shows the pyrolysis temperature dependence of the complex permittivity of these samples in the X-band pyrolyzed at 400–800 °C before heat treatment. The complex permittivity of samples P-400, P-600, and P-800 has no significant change in both real and imaginary parts before the heat treatment. However, the complex permittivity of the SiC/CNTs powders obtained after heat treatment at 400 °C for 200 h in air has changed greatly. For comparison, the real part of permittivity value (ε′) decreases quickly from 13.9 to 6.3, and the imaginary part of permittivity value (ε″) decreases quickly from 5.2 to 0.9 (Figure 5b). When the pyrolysis temperature rises to 400 °C, the pyrolysis of PCS is not complete and cannot form a dense protective layer to block the oxidation of the internal particles. Interestingly, the ε′ and ε″ values of sample P-600 keep almost unchanged in the measured frequency after heat treatment at 400 °C for 200 h (Figure 5c). The oxidation degree of CNTs mainly depends on the PCS decomposition products and the grain size of SiC coating on CNTs surface. In the PCS structure after pyrolyzing at 600 °C, the small molecule oligomers and small molecule gases basically escaped, and the structure was relatively stable. The particles on the surface of the coating had good continuity, while the coating also had a good compactness. For sample P-800, the ε′ and ε″ also decreased, namely 14.6~4.7 and 5.7~0.7 in the X-band (Figure 5d). At 800 °C, as the grains grew, they detached from each other and became more isolated and random, which caused the protective layer to be destroyed. Therefore, the samples of P-400 and P-800 had worse antioxidant properties, more mass loss, and much lower dielectric constant, whereas P-600 had the opposite characteristics. Therefore, it can be demonstrated that sample P-600 has the best oxidation resistance compared with sample P-400 and P-800. Most importantly, it can be deduced that PCS-coated CNTs powders pyrolyzed at 600 °C have excellent dielectric properties in the condition of heat treatment for a long time period.

## 4. Conclusions

In summary, SiC-coated CNTs powders were successfully prepared via the high temperature pyrolysis method. The crystal composition and microstructure of CNTs/SiC materials were confirmed by the results of XRD and SEM. The relative mass loss of PCS and PCS-coated CNTs powders with various heat treatment conditions were characterized via TG-DSC, which confirmed that sample P-600 has a better oxidation resistance at 400 °C for 200 h. In addition, the complex permittivity of CNTs/SiC obtained with different pyrolysis temperatures (400–800 °C) for 0–200 h demonstrated that sample P-600, after heat treatment at 400 °C for 200 h, kept the most excellent dielectric properties. It can be concluded that CNTs/SiC materials can effectively prevent the oxidation of internal particles and maintain the original mass and excellent dielectric properties of particles, which may be usefully applied in high-temperature absorbers.

## Figures and Tables

**Figure 1 materials-14-02770-f001:**
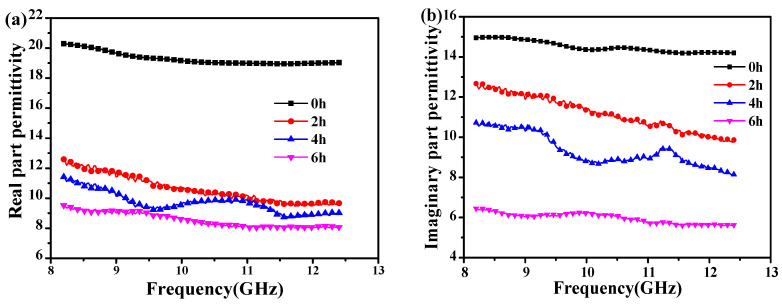
The (**a**) real and (**b**) imaginary part of complex permittivity of CNTs, annealed at 400 °C, with 0–6 h.

**Figure 2 materials-14-02770-f002:**
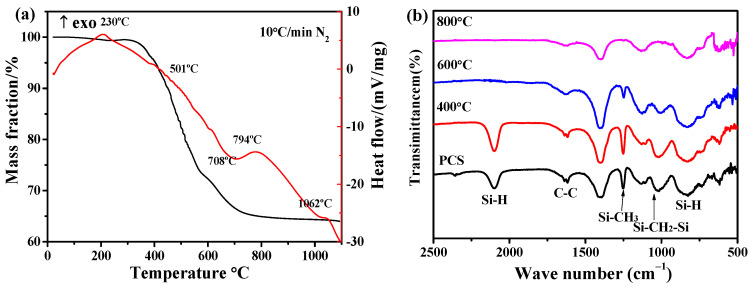
(**a**) TG-DSC curves of PCS, (**b**) FT-IR spectrums of PCS pyrolyzed at different temperatures.

**Figure 3 materials-14-02770-f003:**
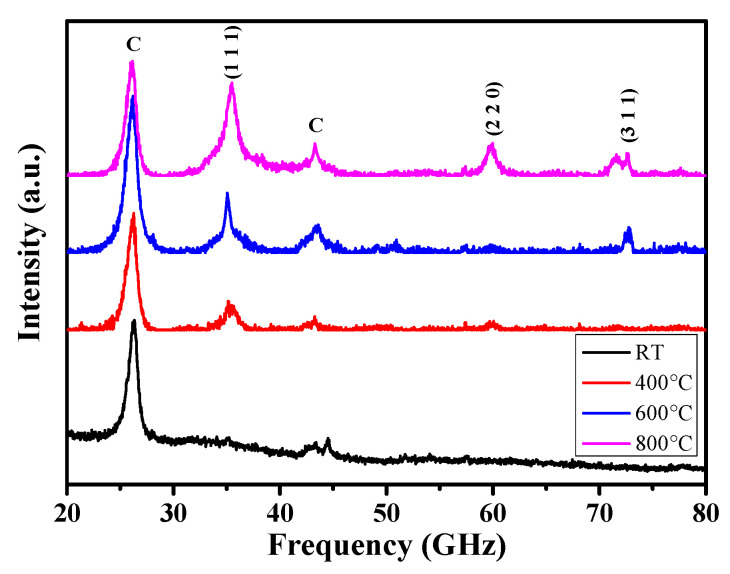
XRD pattern of PCS-coated CNTs pyrolyzed at different temperatures.

**Figure 4 materials-14-02770-f004:**
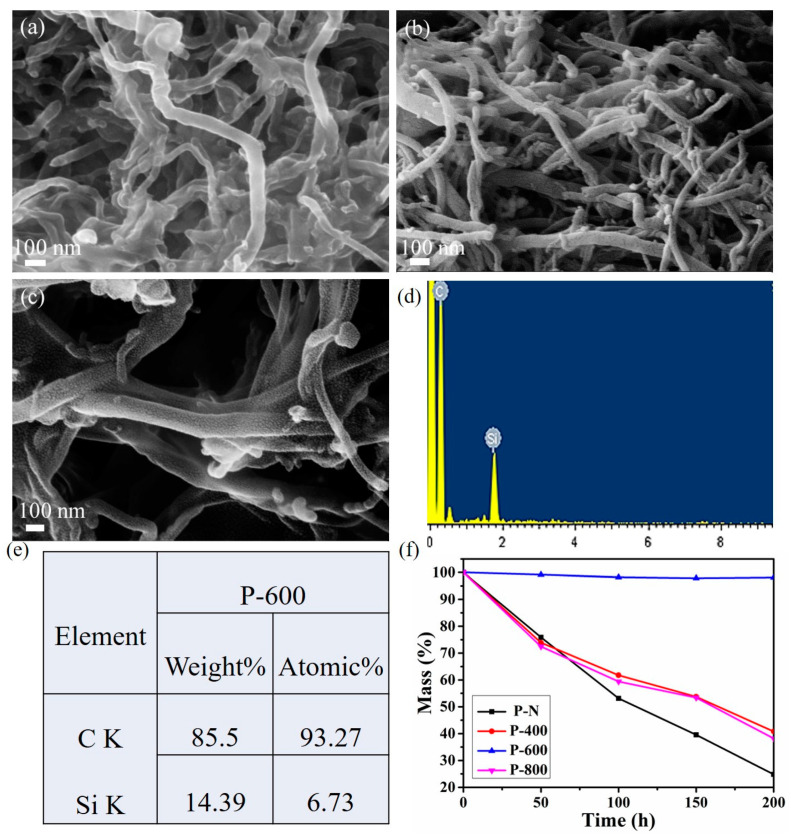
SEM images of PCS-coated CNTs pyrolyzed at different temperatures: (**a**) 400 °C, (**b**) 600 °C, (**c**) 800 °C, (**d**,**e**) EDS of CNTs/SiC, and (**f**) Mass loss of PCS coated CNTs powders pyrolyzed at different temperatures, for 0–200 h.

**Figure 5 materials-14-02770-f005:**
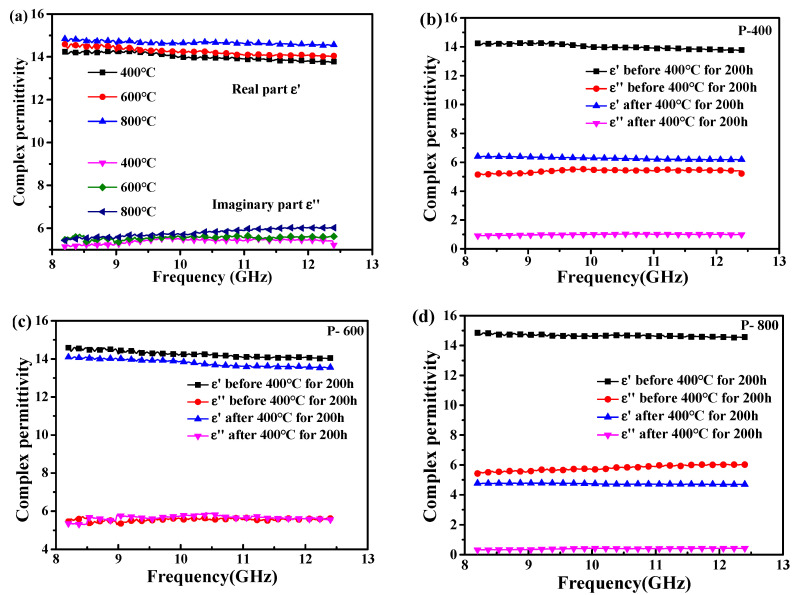
Complex permittivity of (**a**) the PCS-coated CNTs powder pyrolyzed at 400–800 °C before heat treatment and (**b**–**d**) different SiC/CNTs composites after heat treatment at 400 °C in air for 200 h.

## Data Availability

This study did not report any data.

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
