# Peer review of "SiC-Coated Carbon Nanotubes with Enhanced Oxidation Resistance and Stable Dielectric Properties"

_materials, 2021, doi:10.3390/ma14112770_

Round 1

Reviewer 1 Report

The paper signifies the oxidation resistance of SiC coated CNTs; which could be a good material of choice for Electromagnetic wave absorption. The authors have scientifically done the experiments and systematically presented the data. However, the manuscript is not written properly. There are many errors I have highlighted in the attached pdf file, which show the manuscript has to go through major revision before publication. I suggest authors to rewrite the manuscript again and be very careful using abbreviation for samples. For e.g., in the manuscript these 3 phrases “PCS coated CNTs”, “SiC Coated CNTs” and “CNTs/SiC composites” are used for describing the same thing. It makes the reader confused. If written properly, I believe the quality of data and research is good enough to be published in this journal.

Author Response

Responses to the reviewers’ comments

We would like to thank the reviewers for their constructive feedback. Please find below our point-to-point response to each comment. In the following, the reviewer’s comments are in regular font, and our response is italic. In the revised manuscript, uploaded separately, we have addressed all the concerns of the reviewers. The corrected sections are shown in red in the revised manuscript.

Referee: 1
Comments to the Author

The paper signifies the oxidation resistance of SiC coated CNTs; which could be a good material of choice for Electromagnetic wave absorption. The authors have scientifically done the experiments and systematically presented the data. However, the manuscript is not written properly. There are many errors I have highlighted in the attached pdf file, which show the manuscript must go through major revision before publication. I suggest authors to rewrite the manuscript again and be very careful using abbreviation for samples. For e.g., in the manuscript these 3 phrases “PCS coated CNTs”, “SiC Coated CNTs” and “CNTs/SiC composites” are used for describing the same thing. It makes the reader confused. If written properly, I believe the quality of data and research is good enough to be published in this journal.

Authors’ reply: Thank you for your comments. Herein, polycarbosilane (PCS) as a precursor was applicated for the preparation of SiC coatings via high temperature pyrolysis. Before heat treatment, the precursor mixture was called PCS coated CNTs powder. After heat treatment, the obtained product was called SiC Coated CNTs or CNTs/SiC composites. For convenience of distinction, we have checked and relabeled it in this manuscript. Furthermore, numerous grammatical errors indicated on PDF have been also checked and corrected in this manuscript.

Reviewer 2 Report

The present manuscript is an interesting well-written work with actual thematic.
It was interesting for me to read about one of the methods of obtaining SiC coatings for carbon fibers (or CNT like here)
39-40 There is written about SiO2 based protection/ Where the connection with SiC coatings?
68, 70 – the word "merits" should be excluded: “PCS with a softening point…” (do you mean melting?)
83 85 87 89– the font size is different here;
203 Which parameters for the SEM measurements were chosen? a) and b) c) images are looked different. How many experiments were made for (f) measurements Could you repeat it and made a new one for P500/P550/P650 samples? This result is really interesting but here is need more information for the conclusions made. This can significantly improve the experimental validity of the article.
206 –Spaces are absent here.
231 – that is interesting and it correlates with previous data.

Author Response

Responses to the reviewers’ comments

We would like to thank the reviewers for their constructive feedback. Please find below our point-to-point response to each comment. In the following, the reviewer’s comments are in regular font, and our response is italic. In the revised manuscript, uploaded separately, we have addressed all the concerns of the reviewers. The corrected sections are shown in red in the revised manuscript. 

Referee: 2
Comments to the Author

The present manuscript is an interesting well-written work with actual thematic. It was interesting for me to read about one of the methods of obtaining SiC coatings for carbon fibers (or CNT like here)
1. There is written about SiO2 based protection/where the connection with SiC coatings?

Authors’ reply: Thank you for your comments. According to the literature [20-22], the liquid silicon infiltration process is based on the impregnation of composites by molten silicon and its reaction to silicon carbide. A fibre preform with an interconnecting network of cracks is infiltrated by liquid silicon, mostly applying only capillary forces. The application of pressure or the processing in vacuum can improve the infiltration process. The temperatures involved are at least beyond the melting point of silicon (1415℃). In this manuscript, what we actually describe is that the introduction of SiC can effectively provide protection for CNTs.

(References 20-22)

[20] Fan, S.; Zhang, L.; Xu, Y.; Cheng, L.; Lou, J.; Zhang, J.; Yu, L. Microstructure and properties of 3D needle-punched carbon/silicon carbide brake materials. Compos. Sci. Technol. 2007, 67, 2390-2398.

[21] Krenkel, W.; Berndt, F. C/C–SiC composites for space applications and advanced friction systems. Mater. Sci. Eng. A 2005, 412, 177-181.

[22] Xiao, P.; Li, Z.; Zhu, Z.; Xiong, X. Preparation, Properties and Application of C/C-SiC Composites Fabricated by Warm Compacted- in situ Reaction. J Mater. Sci. Technol. 2010, 26, 283-288.

  1. The word "merits" should be excluded: “PCS with a softening point…” (do you mean melting?)

Authors’ reply: Thank you for your comments. The PCS (—[Si-HCH3CH2]n—) with a low melting point (180~210℃), molecular weight (1800~2000) and oxygen content (≤1.0 wt.%) was bought from Suzhou Sailifei Ceramic Fiber Co., Ltd. We have corrected the inappropriate description in this manuscript.

  1. the font size is different here.

Authors’ reply: Thank you for your comments. We have checked and corrected it.

  1. Which parameters for the SEM measurements were chosen? a) and b) c) images are looked different.

Authors’ reply: Thank you for your comments. The scale range of three images in Fig. 4a-c is 100 nm. Due to our negligence, we have revised it in this manuscript.

  1. How many experiments were made for (f) measurements Could you repeat it and made a new one for P500/P550/P650 samples? This result is interesting but here is need more information for the conclusions made. This can significantly improve the experimental validity of the article.

Authors’ reply: Thank you very much for your comments. In this manuscript, we have discussed the oxidation resistance and dielectric properties of P-400, P-600 and P-800 after heat treatment at 400℃ for 200 h. In comparation of the content described in the subtitle 3.1 “Dielectric properties of CNTs annealed at 400℃ with different time”, SiC coated CNTs have a better oxidation resistance and dielectric property in the condition of heat treatment at 400℃ for a long time, which can be deduced that SiC/CNTs composites obtained in the high temperature could be a good candidate for preparing high temperature radar absorbing materials. Based on that, it will be meaningful to investigate oxidation resistance and dielectric properties when the pyrolysis temperature is above 400℃ for 200 h, such as P500, P550 and P650. We will investigate relative experimental result in the next work.

  1. 206-Spaces are absent here.

Authors’ reply: Thank you for your comments. We have checked and corrected it.